# Safety, Tolerability, and Immunogenicity of the Invaplex_AR-Detox_ *Shigella* Vaccine Co-Administered with the dmLT Adjuvant in Dutch and Zambian Adults: Study Protocol for a Multi-Center, Randomized, Double-Blind, Placebo-Controlled, Dose-Escalation Phase Ia/b Clinical Trial

**DOI:** 10.3390/vaccines13010048

**Published:** 2025-01-08

**Authors:** Geert V. T. Roozen, Nsofwa Sukwa, Masuzyo Chirwa, Jessica A. White, Marcus Estrada, Nicole Maier, Kevin R. Turbyfill, Renee M. Laird, Akamol E. Suvarnapunya, Aicha Sayeh, Flavia D’Alessio, Candice Marion, Laura Pattacini, Marie-Astrid Hoogerwerf, Rajagopal Murugan, Manuela Terrinoni, Jan R. Holmgren, Sodiomon B. Sirima, Sophie Houard, Michelo Simuyandi, Meta Roestenberg

**Affiliations:** 1Leiden University Center for Infectious Diseases, Leiden University Medical Center, 2333 ZA Leiden, The Netherlandsl.pattacini@lumc.nl (L.P.); r.a.murugan@lumc.nl (R.M.); 2Centre for Infectious Disease Research Zambia, Lusaka P.O. Box 34681, Zambia; nsofwa.sukwa@cidrz.org (N.S.);; 3PATH, Seattle, WA 98121, USA; jawhite@path.org (J.A.W.); mestrada@path.org (M.E.);; 4Walter Reed Army Institute of Research, Silver Spring, MD 20910, USAakamol.e.suvarnapunya.civ@health.mil (A.E.S.); 5European Vaccine Initiative, 69115 Heidelberg, Germanysophie.houard@euvaccine.eu (S.H.); 6Department of Microbiology and Immunology, University of Gothenburg, 40530 Gothenburg, Sweden; 7Groupe de Recherche Action en Santé, Ouagadougou 06 BP 10248, Burkina Faso

**Keywords:** *Shigella*, shigellosis, diarrheal disease, vaccine, adjuvant, Invaplex, dmLT, clinical trial protocol

## Abstract

Background: Shigella infections remain endemic in places with poor sanitation and are a leading cause of diarrheal mortality globally, as well as a major contributor to gut enteropathy and stunting. There are currently no licensed vaccines for shigellosis but it has been estimated that an effective vaccine could avert 590,000 deaths over a 20-year period. A challenge to effective Shigella vaccine development has been the low immunogenicity and protective efficacy of candidate Shigella vaccines in infants and young children. Additionally, a new vaccine might be less immunogenic in a highly endemic setting compared to a low endemic setting (“vaccine hyporesponsiveness”). The use of a potent adjuvant enhancing both mucosal and systemic immunity might overcome these problems. Invaplex_AR-Detox_ is an injectable Shigella vaccine that uses a novel combination of conserved invasion plasmid antigen proteins and a serotype-specific bacterial lipopolysaccharide attenuated for safe intramuscular administration. The adjuvant dmLT has been shown to enhance Shigella immune responses in mice, has safely been administered intramuscularly, and was shown to enhance immune responses in healthy volunteers when given in combination with other antigens in phase I trials. This article describes the protocol of a study that will be the first to assess the safety, tolerability, and immunogenicity of Invaplex_AR-Detox_ co-administered with dmLT in healthy adults in low-endemic and high-endemic settings. Methods: In a multi-center, randomized, double-blind, and placebo-controlled dose-escalation phase Ia/b trial, the safety, tolerability, and immunogenicity of three intramuscular vaccinations administered 4 weeks apart with 2.5 µg or 10 µg of Invaplex_AR-Detox_ vaccine, alone or in combination with 0.1 µg of the dmLT adjuvant, will first be assessed in a total of 50 healthy Dutch adults (phase Ia) and subsequently in 35 healthy Zambian adults (phase Ib) aged 18–50 years. The primary outcome is safety, and secondary outcomes are humoral and cellular immune responses to the adjuvanted or non-adjuvanted vaccine. Discussion: This trial is part of the ShigaPlexIM project that aims to advance the early clinical development of an injectable Shigella vaccine and to make the vaccine available for late-stage clinical development. This trial addresses the issue of hyporesponsiveness in an early stage of clinical development by testing the vaccine and adjuvant in an endemic setting (Zambia) after the first-in-human administration and the dose-escalation has proven safe and tolerable in a low-endemic setting (Netherlands). Besides strengthening the vaccine pipeline against a major diarrheal disease, another goal of the ShigaPlexIM project is to stimulate capacity building and strengthen global North-South relations in clinical research. Trial registration: EU CT number: 2023-506394-35-02, ClinicalTrials.gov identifier: NCT05961059.

## 1. Introduction

Shigellosis is an acute invasive enteric infection caused by the gram-negative, non-motile *Shigella* bacillus. *S. flexneri* is the main serogroup found in low-income countries (~60% of isolates), followed by *S. sonnei* (~15–25%) [1,2]. In middle- and high-income countries, *S. sonnei* is most common (~60%), followed by *S. flexneri* (~15–20%) [3,4].

*Shigella* is transmitted through the fecal-oral route, and its infective dose is very low. This gives rise to outbreaks in situations with poor sanitation, enabling the disease to disseminate rapidly by person-to-person contact [3]. Shigellosis clinically manifests after an incubation period of 1 to 4 days with symptoms of fever, malaise, anorexia, and sometimes vomiting. *Shigella* infection can cause bloody diarrhea but in mild infections, watery diarrhea can be the only clinical manifestation [3]. Additionally, local invasion and intestinal inflammation induced by *Shigella* infection are risk factors for the subsequent development of gut enteropathy, malnutrition, and stunting in children [5,6]. Shigellosis is a leading cause of diarrheal deaths globally [3] and is the fourth most common cause of moderate to severe diarrhea in young children living in low- and middle-income countries (LMICs) [7]. In 2016, shigellosis caused over 200,000 deaths of which more than 60,000 were children under 5 years old [8]. A recent systematic review found *Shigella* to be the second leading cause of diarrhea-associated mortality in LMICs [9].

There is currently no licensed vaccine available for shigellosis [10]. Following a rise in antibiotic resistance combined with the high burden of infection, particularly in LMICs, the World Health Organization (WHO) has identified *Shigella* as a priority bacterial pathogen and has urged the development of a *Shigella* vaccine [11,12,13]. It has been estimated that *Shigella* vaccination could avert 43 million cases of stunting and 590,000 deaths over a 20-year period and would be highly cost-efficient [14,15].

Invaplex_AR-Detox_ is a subunit *Shigella* vaccine that contains conserved invasion plasmid antigen (Ipa) proteins (IpaB and IpaC) together with an *S. flexneri* 2a serotype-specific bacterial lipopolysaccharide (LPS). In addition to *S. flexneri* 2a, the recombinantly-produced conserved proteins IpaB and IpaC could also offer protection against *S. flexneri* serotypes 3 and 6, and *S. sonnei* [16]. The LPS is generated from a genetically attenuated msbB *Shigella* mutant that expresses underacylated and therefore detoxified lipid A, thus enabling safe intramuscular (IM) vaccine administration [17]. Previously, a phase I trial (NCT03869333) was conducted in 58 adults in the United States, assessing three different dose levels (2.5, 10, and 25 μg) of Invaplex_AR-Detox_ given IM in a three-dose course. All three dose levels of Invaplex_AR-Detox_ were well tolerated and highly immunogenic, inducing strong systemic antibody responses to all three *Shigella* antigens contained within the vaccine. The magnitude of the LPS-specific serum IgG responses was comparable to or exceeded antibody levels observed after infection with *S. flexneri* 2a [18,19]. Antibody responses to the IpaB and IpaC antigens induced by the vaccine also approached levels that have been associated with protection against shigellosis [20]. Although these results are promising in the adult population, historically, reduced immunogenicity and protective efficacy of *Shigella* vaccines have been seen in infants and thus provide a rationale for adding an adjuvant to enhance immunogenicity [12,21].

A promising adjuvant candidate is the double mutant (LT R192G/L211A) enterotoxigenic *E. coli* heat-labile toxin (dmLT) [22] that has previously been shown to enhance both systemic and mucosal immune responses to *Shigella* antigens in mice [23]. Furthermore, dmLT (up to 0.5 µg) has been administered IM in a phase I trial of a prototype enterotoxigenic *E. coli* vaccine antigen and was found to be safe, and significantly improved antibody responses to this vaccine antigen [24].

This study will be the first clinical trial assessing the safety, tolerability, and immunogenicity of the Invaplex_AR-Detox_ vaccine adjuvanted with dmLT. The first-in-human and dose-escalation part of the trial will be conducted in The Netherlands, in a setting with low *Shigella* endemicity, and therefore with immunologically naïve participants. The second part of the trial will take place in an endemic setting in Zambia.

## 2. Materials and Methods

### 2.1. Design

This will be a multi-center, randomized, double-blind, placebo-controlled, dose-escalation phase Ia/b clinical trial assessing the safety, tolerability, and immunogenicity of three IM vaccinations given 28 (±5) days apart. The vaccine will consist of either a 2.5 or 10 µg dose of the Invaplex_AR-Detox_ vaccine administered alone or in combination with a 0.1 µg dose of the dmLT adjuvant and will be compared to placebo (saline solution). A total of 85 healthy participants aged 18 to 50 years will be recruited: 50 in The Netherlands and 35 in Zambia. Inclusion and exclusion criteria are reported in Table 1.

In the first part of the trial (phase Ia), which will include the dose-escalation evaluation, the participants will be enrolled in two cohorts (A and B) at the Leiden University Medical Center (LUMC) in Leiden, The Netherlands (Figure 1). In the second part of the trial (phase Ib), the participants will be enrolled in a single cohort (C) at the Centre for Infectious Disease Research in Zambia (CIDRZ) in Lusaka, Zambia.

In cohort A, a total of 25 participants will receive 2.5 µg Invaplex_AR-Detox_ with or without 0.1 µg dmLT, or a placebo in a 2:2:1 ratio, respectively (Table 2). In cohort B, a total of 25 participants will receive 10 µg Invaplex_AR-Detox_ with or without 0.1 µg dmLT, or a placebo in a 2:2:1 ratio, respectively. In cohort C, a total of 35 participants will receive 10 µg Invaplex_AR-Detox_ with or without 0.1 µg dmLT, or a placebo in a 3:3:1 ratio, respectively. As an additional safety measure, all three cohorts will be divided into a sentinel cohort and a remaining cohort. A sentinel cohort consists of seven participants (three receiving the vaccine alone, three receiving the adjuvanted vaccine, and one participant receiving a placebo). The seven participants in a sentinel cohort will be vaccinated on the same day with a 30-min interval. Only if the vaccine and adjuvant are safe and well tolerated in the sentinel cohort in the 7 days following administration will the remaining participants of the cohort receive their dose. At both sites, an independent local safety officer will monitor trial progress and safety and will decide together with the local principal investigator (PI) if the remainder of the cohort will be vaccinated.

Participants will register adverse events (AEs) and body temperature on a daily basis in a paper diary. They will return for a study visit 7 days post-vaccination for the collection of samples and a safety assessment. In Zambia, investigators will perform home visits in the week after each vaccination to ensure adequate recording of AEs and body temperature. After the third vaccination, there will be a visit at 7 and 28 days after vaccination. Twenty-four weeks after the third vaccination, the last visit will take place. The decision to proceed from one cohort to the next will be based on the recommendation of the safety monitoring committee (SMC). The SMC will consist of two local safety officers with two additional clinicians: a pediatrician and an expert in vaccines for gastrointestinal infectious diseases. The local safety officer may be an in-house expert at a study site, but not a member of the study team. The two additional SMC members will have no affiliation to the study sites. One of the independent SMC members will be appointed as chair and has a casting vote in case of a tie. If the 10 µg Invaplex_AR-Detox_ dosage is not well tolerated in cohort B or deemed unsafe by the SMC, the committee may recommend proceeding to cohort C with the 2.5 µg Invaplex_AR-Detox_ dosage.

### 2.2. Participants

At the LUMC in The Netherlands, recruitment of healthy volunteers from the general population will be performed through advertisements, social media, and institutional websites. Advertisement material will be subject to ethical review before the start of recruitment. At CIDRZ in Zambia, volunteers will be recruited from Matero General Hospital, CIDRZ clinical research site, and the surrounding community. Individuals should be able to understand and comply with planned study visits and procedures. After a detailed explanation of the trial, participants will have to provide written informed consent prior to any of the trial procedures. Pregnant or breastfeeding women, individuals who recently had gastroenteritis, and individuals with an abnormal stool pattern or regular users of antidiarrheal, anti-constipation, or antiacid therapy are excluded. Women of childbearing potential must agree to use continuous highly effective contraception to avoid pregnancy during the trial, for at least 4 weeks before the first vaccination and for 3 months following the last vaccine dose.

### 2.3. Screening

During a screening visit, information on demographics, self-reported medical and medication history, and alcohol, drugs, and cigarette use will be recorded. A general physical examination including vital sign measurements will be performed and samples will be collected for laboratory tests including hematology (e.g., hemoglobin, red blood cell count, platelet count), serum chemistry (e.g., creatinine, sodium, potassium, blood urea nitrogen, transaminases), serology (human immunodeficiency virus, hepatitis B and C) and a diagnostic polymerase chain reaction (PCR) for *Shigella* on a stool sample. Women of childbearing potential will also be subjected to a urine or serum pregnancy test, both at screening and before each vaccination. Individuals with a clinically significant abnormal test result or a positive *Shigella* PCR will be excluded from the trial.

### 2.4. Randomization and Blinding

Before the first vaccination in a sentinel or remaining cohort takes place, a randomization list will be prepared by at least two members of the pharmacy team not otherwise involved in the trial. The allocation ratios will be 3:3:1 for all sentinel cohorts, 7:7:4 for remaining cohorts A and B, and 3:3:1 for remaining cohort C (Table 2). The randomization list will link each randomization number to the corresponding intervention (vaccine alone, adjuvanted vaccine, or placebo). The pharmacy team will prepare the investigational products and will assign the corresponding randomization number to each product. Investigators will receive the investigational product labeled with a randomization number. The investigator will administer the investigational product to the participant and will record the corresponding randomization number for the said participant in the electronic case report form (eCRF). This participant-randomization number combination will be communicated to the pharmacy team to allocate the correct investigational product to the participant during the subsequent administrations. Both investigators and participants will be blinded to the intervention. Each participant will remain blinded until the end of the trial. After data cleaning, the database will be locked, followed by unblinding of the cohorts.

### 2.5. Vaccination

Vaccination will be done through IM injection. The first vaccine dose will be administered in the left deltoid, the second in the right deltoid, and the third in the left deltoid, to ensure that local side effects can be correctly attributed to the corresponding administration. In sentinel cohorts, there will be at least 30 min between the vaccination of subsequent participants. In the remaining cohorts, this delay will be at least 15 min. After vaccination, participants will be observed for at least 30 min to ensure neither hypersensitivity nor anaphylactic reaction occurs.

### 2.6. Outcomes

The primary outcome is safety. The safety evaluation will include registration of all solicited and unsolicited AEs during the whole trial period (Table 3). All reported AEs will be classified according to the International Classification of Diseases version 11 (ICD-11) and causality and severity will be assessed by investigators (Table 3). Primary outcome measures will consist of all solicited AEs considered to be possibly, probably, or definitely related to vaccination occurring within 7 days following each vaccination and all unsolicited AEs occurring within 28 days following each vaccination. There are three pre-specified safety-stopping rules (Table 4). If a stopping rule is met, the study is put on a temporary halt and no further vaccines will be administered until the temporary halt is suspended. AE collection and blood collection will still continue. If not done already, the hospital pharmacist will break the study blind for the affected volunteers and the SMC will be informed about the group allocation. The trial participants and all investigators will remain blinded. The SMC will review the safety data and recommend the study sponsor on study (dis)continuation.

The secondary outcome is immunogenicity. To assess serum anti-LPS, anti-IpaB, and anti-IpaC IgG and IgA antibody responses and to perform serum bactericidal assays (SBA) to *S. flexneri* 2a (strain 2457T), serum samples will be collected at days 1 (baseline), 29, 57, 64, 85, and 225 and kept frozen at −20 °C until tested. Peripheral blood mononuclear cells will be collected and cryopreserved at days 1, 8, 64, and 225 to assess B and T cell responses to the LPS, IpaB, and IpaC antigens. Further exploratory humoral and cellular immune assays using the cryopreserved serum and mononuclear cell samples may be performed as resources allow.

### 2.7. Data Management and Monitoring

An eCRF will be used at both sites to facilitate direct data entry by the investigators for visit reports and AE registration. Other documentation (e.g., paper diaries, laboratory reports, or supplementary medical records) may form part of the source documentation for a study participant. These source documents will be filed and stored at the site where they were produced. To create the eCRF, a validated system with audit, data, and edit trail will be used to comply with Good Clinical Practice (GCP) standards for electronic data entry and export. Both sites will be monitored by an independent monitor to ensure the study will be conducted in accordance with the study protocol and GCP standards.

### 2.8. Statistical Analysis and Sample Size

Safety will be assessed in the intention-to-treat (ITT) population. The ITT population will be defined as all participants included in the trial (i.e., all participants who received at least one vaccine dose). Safety data will be grouped by study cohorts A, B, and C, except for control participants who received the placebo, and will be grouped together. AEs will be reported as both frequencies and percentages. Rates will be compared by Pearson’s Chi-square test (or Fisher’s exact test if assumptions are not met for Pearson’s Chi-square).

Immunological analyses will be performed on the per-protocol (PP) population. The PP population will be adapted to each analysis time point and parameter and will include all participants who received at least two vaccinations and have baseline and postvaccination data for the immunogenicity variable of interest. Participants with missing baseline data, or major protocol deviations prior to the analysis time point that are likely to affect immunology results will be excluded. Analyses of immunological data will be grouped by cohorts A, B, and C, except for control participants who received the placebo and will be grouped together. Additionally, an analysis per dose level (2.5 and 10 µg) may be performed.

Reciprocal endpoint titers less than the starting dilution (the lowest limit of quantification) of the assay will be assigned a value of half the starting dilution for computational purposes. In general, descriptive statistics (mean and SD of log_10_ titers, geometric mean titer [GMT] and 95% CI, median, range) will be tabulated by cohort and time point. The two-sided 95% CI will be obtained using a t-distribution. Additionally, the geometric mean fold-rise from baseline (GMFR) will be computed (based on the difference in log titer of post-baseline measurement minus baseline) and summarized in the same manner. For GMTs and GMFRs, between-group comparisons will be examined with ANOVA. The normality of the log-transformed continuous outcomes will be assessed using goodness-of-fit tests based on the empirical distribution function and by inspection of the normal probability plot. If normality assumptions are not satisfied, then the Kruskal-Wallis test will be used.

The number and proportion of responders (participants who seroconvert with a ≥4-fold increase in endpoint titer between baseline and post-vaccination samples), together with exact Clopper-Pearson 95% CIs will be tabulated by cohort and time point. If appropriate, between-group comparisons will be examined with Fisher’s exact test unless assumptions are fulfilled for the χ^2^ test.

A sample size of 10 participants per intervention group will allow for a reasonable initial assessment of the AE profiles of the first-in-human administration of the adjuvanted vaccine (cohort A) and the subsequent dose escalation (cohort B). A group size of 15 participants in cohort C will provide 80% power to detect AEs with an assumed true event rate of 10% in each intervention group separately.

## 3. Discussion

This trial is part of the ShigaPlexIM project (www.shigaplexim.eu, accessed on 13 December 2024), a collaborative project funded by the European and Developing Countries Trials Partnership 2 (EDCTP2). ShigaPlexIM aims to advance the early clinical development of an adjuvanted injectable *Shigella* vaccine and to make the vaccine available for late-stage clinical development. Prior to the clinical trial, data was collected in a surveillance study on the incidence of shigellosis in Burkina Faso and Zambia among children under 5 presenting with moderate to severe diarrhea at primary healthcare facilities to provide an epidemiologic basis for further clinical development and late-stage clinical trial design (manuscript in preparation).

In a previous trial (NCT03869333), a three-dose regimen of Invaplex_AR-Detox_ was safe and immunogenic in US adults, inducing high anti-LPS, anti-IpaB, and anti-IpaC antibody titers. Previous vaccines have shown to be less immunogenic in endemic settings due to genetic differences and differences in environmental factors such as food intake, microbiome, and exposure to different micro-organisms and parasites; a phenomenon called vaccine hyporesponsiveness [25]. The trial described here aims to address the issue of hyporesponsiveness in an early stage of clinical development by testing the Invaplex_AR-Detox_ vaccine in combination with an adjuvant in a highly endemic setting after the first-in-human administration and the dose-escalation has proven safe and tolerable in a low endemic setting.

Even though the protective immune mechanisms against shigellosis are not fully understood, antibodies against the cell wall LPS antigen are believed to play a primary role, with additional protective effects from an immune response against virulence proteins IpaA, B, C, and D and VirG [20,26,27]. The current formulation of Invaplex_AR-Detox_ should induce immune responses to *S. flexneri* serotypes 2a, 3a, and 6. LPS from *S. sonnei* will be added at a later stage of product development to optimize serotype coverage: *S. flexneri* 2a, 3a, and 6, and *S. sonnei* together cover about 80% of the strains causing shigellosis [2].

As defined by the WHO, the primary target population for a new *Shigella* vaccine consists of infants and young children [12,13]. After the current trial in adults, the intention is to conduct a phase IIa age de-escalation study in Zambia and/or Burkina Faso to test the vaccine and adjuvant sequentially in adolescents, young children, and infants. The data on the safety, tolerability, and immunogenicity of the current trial in adults will enable an informed decision on proceeding to a phase IIa study and will provide the foundation for the design of such a trial. An age de-escalation study in an African setting will answer the question if Invaplex_AR-Detox_ with or without dmLT will be able to overcome problems of low immunogenicity in the target population that have hampered the clinical development and implementation of *Shigella* vaccines in the past [12,21].

Besides strengthening the vaccine pipeline against a major diarrheal disease, another goal of the ShigaPlexIM project is to stimulate capacity building and to provide support for formal education and training of study staff. The close collaboration will develop capacity in clinical trial conduct and clinical immunology and will strengthen both global South-South and global North-South networks. To further strengthen these connections, the creation of new PhD positions and dedicated capacity-building events are separate deliverables of the ShigaPlexIM project.

## Figures and Tables

**Figure 1 vaccines-13-00048-f001:**
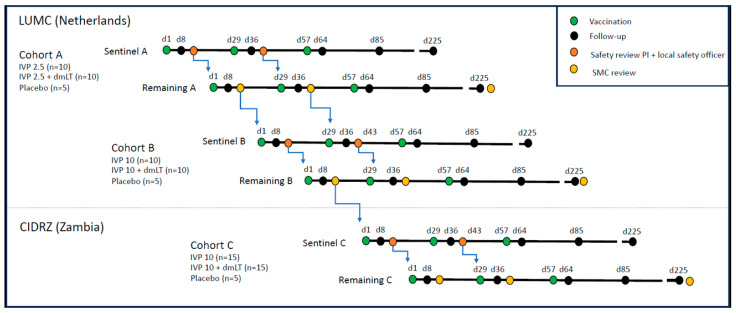
Study design. Participants are vaccinated on days 1, 29, and 57. In each sentinel cohort, a safety review by the PI and a local safety officer will take place after the first vaccination (between days 8 and 29) and the second vaccination (between days 36 and 57). In the remaining cohorts, an SMC review will take place on these same time points. Only after these reviews have found the vaccine and adjuvant to be safe and well-tolerable, can further vaccination of the subsequent cohort take place (indicated by blue arrows). CIDRZ: Center for Infectious Disease Research in Zambia; IVP 2.5: 2.5 µg Invaplex_AR-Detox_; IVP 10: 10 µg Invaplex_AR-Detox_; LUMC: Leiden University Medical Center; PI: principal investigator; SMC: safety monitoring committee.

**Table 1 vaccines-13-00048-t001:** Inclusion and exclusion criteria.

Inclusion criteria	- Healthy adult, male or female, aged 18 to 50 years (inclusive) at the time of inclusion (=first vaccination). - Provide written informed consent before initiation of any study procedure. - Available to complete all study visits and procedures. - Negative stool PCR test for *Shigella.* - Women of childbearing potential: negative pregnancy test at screening and before each study vaccine administration. Women of childbearing potential must agree to use continuous highly effective contraception to avoid pregnancy during the study, starting at least 4 weeks before the first vaccine dose, until 3 months following the last vaccine dose.
Exclusion criteria	- Any history or evidence of clinically relevant chronic medical conditions (such as psychiatric conditions, diabetes mellitus, hypertension [treated by medication], autoimmune disorders, immunodeficiencies, cardiovascular disease, renal disease, or inflammatory bowel disease). Trial physicians (in consultation with the principal investigator) will use clinical judgment on a case-by-case basis to assess safety risks under this criterion. - Current use of immunosuppressive medications (except for antihistamines and topical or inhalation corticosteroids). - Women who are a) currently nursing or b) who are pregnant or planning to become pregnant during the study period plus 3 months beyond the last vaccine dose. - Participation in research involving another investigational product (defined as receipt of an investigational product or exposure to an invasive investigational device) 30 days before the first vaccination or anytime through the last in-clinic study safety visit. - Positive blood test for hepatitis B surface antigen (HBsAg), hepatitis C virus (HCV), or human immunodeficiency virus (HIV). - Clinically significant abnormalities on basic laboratory screening tests. - Systemic antimicrobial treatment (i.e., topical treatments are not an exclusion criterion) within 1 week before the first vaccine dose (temporary exclusion). - Known hypersensitivity to compounds in the vaccine or adjuvant or other known drug allergies that may increase the risk of adverse events. - Regular use (weekly or more often) of antidiarrheal, anti-constipation, or antacid therapy. - Abnormal stool pattern (fewer than 3 stools per week or more than 3 stools per day) on a regular basis; loose or liquid stools on other than an occasional basis. - Personal or family history of inflammatory arthritis. - Proven allergy to any substance in the Invaplex_AR-Detox_ vaccine or dmLT or history of anaphylactic reaction to any other vaccine. - Exclusionary skin disease history/findings that would confound assessment or prevent appropriate monitoring of local adverse events, or possibly increase the risk of local adverse events. - Recent (<3 months) history of gastroenteritis. - Received previous licensed or experimental *Shigella* vaccine, dmLT, or live *Shigella* challenge. - Any severe medical condition that might place the participant at increased risk of adverse events according to the clinical judgment of the study clinicians in consultation with the PI. - Any planned vaccination within 14 days before the first vaccine dose until the last in-clinic visit, with the exception of SARS-CoV-2 vaccines or influenza vaccines.

**Table 2 vaccines-13-00048-t002:** Trial sites, cohorts, and groups with corresponding vaccine and adjuvant doses.

Site	Cohort	Group	Invaplex_AR-Detox_ (µg)	dmLT (µg)	N
LUMC	A	A1	2.5	---	10
		A2	2.5	0.1	10
		A3	Placebo	---	5
		- Sentinel cohort (A1: n = 3, A2: n = 3, A3: n = 1) to be dosed first, followed by the remaining cohort a week later. - Advancement to cohort B based on safety data through 7 days post first dose.			
	B	B1	10	---	10
		B2	10	0.1	10
		B3	Placebo	---	5
		- Sentinel cohort (B1: n = 3, B2: n = 3, B3: n = 1) to be dosed first, followed by the remaining cohort a week later. - Advancement to cohort C based on safety data through 7 days post first dose.			
CIDRZ	C	C1	10 *	---	15
		C2	10 *	0.1	15
		C3	Placebo	---	5
		- Sentinel cohort (C1: n = 3, C2: n = 3, C3: n = 1) to be dosed first, followed by the remaining cohort a week later. - Advancement to age-descending study (phase IIa) based on safety data from cohorts A and B (LUMC) through 6 months and safety data from cohort C through 7 days post third dose.			

* In case the 10 µg dose is not well tolerated or deemed unsafe by the safety monitoring committee, the committee can decide to proceed to cohort C with the 2.5 µg dose.

**Table 3 vaccines-13-00048-t003:** Adverse events classification.

Solicited *	
Local	Pain/tenderness, erythema, induration/swelling, pruritus, ipsilateral axillary lymphadenopathy, and depigmentation.
Systemic	Fever, chills, headache, fatigue, malaise, nausea/vomiting, painful/swollen joints, myalgia, diarrhea, abdominal pain, and neuralgia.
**Severity**	
Mild	Awareness of symptoms that are easily tolerated and do not interfere with usual daily activity.
Moderate	Discomfort that interferes with or limits usual daily activity but does not require professional medical attention.
Severe	Disabling, with subsequent inability to perform the usual daily activity, resulting in absence from work or school, requiring bed rest during the majority of the day, and may need professional medical attention.
Serious	Requires emergency room visit or hospitalization or results in persistent or significant disability, incapacity, or death.
**Relatedness**	
Not related	A relationship to the administration of the investigational product cannot be reasonably established or another etiology is known to have caused the adverse event or is highly likely to have caused it.
Unlikely	A relationship to the administration of the investigational product is unlikely; however, it cannot be ruled out.
Possibly	There is a potential association between the event and administration of the investigational product; however, there is an alternative etiology that is more likely.
Probably	Administration of the investigational product is the most likely cause; however, there are alternative reasonable explanations, even though less likely.
Definitely	Administration of the investigational product is the only reasonable cause; another etiology causing the adverse event is not known.

* Only symptoms with an onset after a vaccination until the 7th subsequent day after that vaccination will be considered solicited.

**Table 4 vaccines-13-00048-t004:** Pre-specified stopping rules.

A serious adverse event occurs that is possibly, probably, or definitely related to InvaplexAR-Detox and/or dmLT. In case of a blinded assessment: if more than 25% of participants in one cohort (i.e., sentinel cohort and remaining cohort combined) experience one or more severe adverse events that are possibly, probably, or definitely related to vaccination and last more than 48 h. In case of an unblinded assessment: if more than 50% of participants in one dosing group experience one or more severe adverse events that are possibly, probably, or definitely related to vaccination and last more than 48 h.

## Data Availability

Data sharing is not applicable to this article as no datasets were generated or analyzed yet during this study.

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
