# Peer review of "Safety, Tolerability, and Immunogenicity of the InvaplexAR-Detox Shigella Vaccine Co-Administered with the dmLT Adjuvant in Dutch and Zambian Adults: Study Protocol for a Multi-Center, Randomized, Double-Blind, Placebo-Controlled, Dose-Escalation Phase Ia/b Clinical Trial"

_vaccines, 2025, doi:10.3390/vaccines13010048_

Round 1

Reviewer 1 Report

Comments and Suggestions for Authors

Overall, well designed study with very relevant topic, much needed vaccine.

Few suggestions/comments for improvement/clarification.

1. Page 5, Lines 134-5

Please define sentinel cohort, all participants will be recruited at the same time or sequentially one by one with 24/48 hr gap?

2. Page 5, Lines 149-53

-Please clarify if there will be pre-defined study halting rules in the protocol or not.

-Also please clarify if SMC is independent of Sponsor & site or in house expertise?

-Generally, SMC/DSMB members are odd numbers, in case of tie, odd numbers helps in decision making

3. Page 5, Lines 156-7

Please clarify if advertisement material will be approved by site IRB?

4. Page 6, Line 198

Rationale for 30 min please? Generally 24 hr gap is used for sentinel requirement

5. Page 7  ,Line 215

In Zambia, during the study duration, any diarrhea cases will be cultured/PCR for Shigella as exploratory?

6. Page 8,  Line 236

Protocol defined vaccine schedule is 3 doses, so how 2 doses will be considered on PP analysis?

7. Page 8, Lines 253-4

Please mention LLOQ of the assay, how would you handle values below LLOQ?

8. Page 8, Lines 258-62

Is there any criteria defined for dose selection based on Immune results? or same will be done in phase 2 in children?

9. Author by consider following additional exclusion criteria if study is not already ongoing

Febrile illness (temperature ≥ 38°C) within the 3 days of intended study vaccination

Individuals who have received blood, blood products, and/or plasma derivatives including parenteral immunoglobulin preparations in the past 12 weeks.

Individuals with Body Mass Index (BMI)> 30 kg/m2

Individuals with history of substance or alcohol abuse within the past 2 years.

Individuals who have a previously laboratory confirmed or suspected disease caused by Shigella. 

Individuals who have had household contact with/and or intimate exposure to an individual with laboratory confirmed Shigella. 

Reviewer 2 Report

Comments and Suggestions for Authors

Roozen et al conducted a study protocol for a multi-center, randomized, double-blind, placebo-controlled dose-escalation phase Ia/b clinical trial in order to study the safety, tolerability and immunogenicity of the InvaplexAR-Detox Shigella vaccine co-administered with the dmLT adjuvant in Dutch and Zambian adults. The manuscript follows a logical order and the topic of the manuscript is of interest. The content of the manuscript is well structured and clearly presents the information in a meaningful way to the reader. I just have three minor points, which are detailed below and should be addressed before publication. 

1- Figure 1: Please, include in the legend the meaning of “AI”, which day the “safety review PI” and “SMC review” will be conducted, and what blue arrows indicate.

2- Lines 181-2: “7:7:4 for remaining cohorts A and B”. Please, clarify if this ratio corresponds to phase IIa, because it is not indicated in table 2.

3- Lines 238-9: “ major protocol deviations prior to the analysis time point that are likely to affect immunology results will be excluded” Could you detail which “protocol deviations” you will consider to exclude?
